# Emerging Resistance of Neglected Tropical Diseases: A Scoping Review of the Literature

**DOI:** 10.3390/ijerph16111925

**Published:** 2019-05-31

**Authors:** Folahanmi T. Akinsolu, Priscilla O. Nemieboka, Diana W. Njuguna, Makafui N. Ahadji, Dora Dezso, Orsolya Varga

**Affiliations:** Department of Preventive Medicine, Faculty of Public Health, University of Debrecen, H-4002 Debrecen, Hungary; datavivida@yahoo.com (P.O.N.); diana.njuguna@sph.unideb.hu (D.W.N); makafuinuna@gmail.com (M.N.A.); dezso.dora@sph.unideb.hu (D.D.); varga.orsolya@sph.unideb.hu (O.V.)

**Keywords:** antimicrobial resistance, drug resistance, monitoring, neglected tropical diseases, surveillance

## Abstract

*Background:* Antimicrobial resistance (AMR) is a global public health threat with the potential to cause millions of deaths. There has been a tremendous increase in the use of antimicrobials, stemming from preventive chemotherapy elimination and control programs addressing neglected tropical diseases (NTDs). This study aims to identify the frequency of drug resistance for 11 major NTDs and 20 treatment drugs within a specific period by systematically analyzing the study design, socio-demographic factors, resistance, and countries of relevant studies. *Methods*: Adhering to PRISMA guidelines, we performed systematic reviews of the major 11 NTDs to identify publications on drug resistance between 2000 and 2016. A quality assessment tool adapted for evaluating observational and experimental studies was applied to assess the quality of eligible studies. *Results:* One of the major findings is that six NTDs have information on drug resistance, namely human African trypanosomiasis, leishmaniasis, onchocerciasis, schistosomiasis, soil-transmitted helminths, and trachoma. Many studies recorded resistance due to diagnostic tests, and few studies indicated clinical resistance. Although most studies were performed in Africa where there is the occurrence of several NTDs, there was no link between disease burden and locations of study. *Conclusions:* Based on this study we deduce that monitoring and surveillance systems need to be strengthened to enable the early detection of AMR and the mitigation of its global spread.

## 1. Introduction

In the past two decades, over two billion of the world’s poorest people have been affected by neglected tropical diseases (NTDs). NTDs are mainly grouped into parasitic, viral, and bacterial infections in Africa, Asia, and America [1,2]. The widespread and often catastrophic consequences of NTDs necessitates a global response, prompting organizations such as the United States Centers for Disease Control and Prevention (CDC), the United Nations (UN), and the World Health Organization (WHO) to focus on them. The WHO has identified 20 NTDs [3,4], out of which 11 are considered as major NTDs [5]. The 11 major NTDs studied are Chagas disease, food-borne trematodiasis, human African trypanosomiasis (HAT), leishmaniasis, leprosy, lymphatic filariasis, onchocerciasis schistosomiasis, soil-transmitted helminths (STH), taeniasis, and trachoma.

The categorization of these diseases as “neglected” was established by Peter Hotez, Alan Fenwick, and Alan Fairlamb after the establishment of the Millennium Development Goals (2000) [6]. The WHO launched its first report on NTDs in 2010, which defined the strategic approaches for reducing the burden of identified diseases [7], and provided a “roadmap” revealing the 2015 and 2020 targets for eradication, elimination, and intensified control. 

The two primary methods of interventions for NTDs are preventive chemotherapy and transmission control (PCT), which covers mass drug administration (MDA) and innovative and intensified disease management (IDM) [2]. Global strategies and applicable tools are readily available for PCT [2]. IDM focuses more on NTDs, for which simple tools and treatments are not yet available and wide-scale prevention cannot be applied [2,8]. According to the WHO, an estimate of 1.7 billion people in 185 countries needed mass and/or individual treatment and care for NTDs in 2014 [9]. In recent times, tremendous steps have been taken to curtail NTDs by the Global Network and public–private partnerships. In 2011, there was a 37% average coverage of PCT for NTDs, but with the involvement of strong partnerships, the average coverage of PCT increased to 63% in 2016 [10]. PCT-covering MDA programs have been considered to be effective, in spite of the potential for the development of drug resistance due to long and continuous usage, which remains a challenge [11]. However, the overall progress has been uneven. According to Hotez, the WHO has discovered that less than two-thirds of the global population that needs treatment for NTDs are covered, and the treatment for trachoma and schistosomiasis is quite inadequate [4,12].

Antimicrobial resistance (AMR) is a global public health threat, and its impacts have the potential to kill millions of people. It is also a fundamental commercial challenge for private sector companies because developing new antimicrobials is often expensive and requires long-term proposition [9]. In recent times, AMR has increasingly become a problem because of a tremendous increase in the use of antimicrobials, which has caused an increase in the rate at which resistance is developing and spreading. Unfortunately, there are no adequate new antimicrobials to address this situation [13]. The WHO is committed to developing a global consensus approach to AMR monitoring, with predefined measures of impact and outcome consistent with the Global Action Plan (GAP) [14]. The Global Antimicrobial Resistance Surveillance System provides a cornerstone for assessing the spread of AMR, by informing and monitoring the impact of local, national, and global strategies [15]. Evaluating the impact of implemented programs and interventions towards the prevention and control of tropical diseases would be effective, with a surveillance-response system implemented to achieve the maximum amount of positive health outcomes [16].

Importantly, AMR does not recognize geographic or human/animal species borders. Addressing the rising threat of AMR requires the “One Health” approach, which addresses human health, animal health, and the environment. Although the WHO, the Food and Agriculture Organization, and the World Organization for Animal Health have all taken collective action to minimize the emergence and spread of AMR through the “Tripartite Collaboration”, there are still limitations with the agreement [17,18]. Collective action is required in areas of surveillance, infection control, awareness, and responsible use for successful containment of AMR emergence and spread.

The aim of this study was to identify the frequency of resistance for 11 major NTDs and 20 drugs over a specific period by systematically analyzing the study design, socio-demographic factors, resistance, and the countries of the studies.

## 2. Methods

### 2.1. Protocol Registration

This study was performed according to the Preferred Reporting Items for Systematic Reviews (PRISMA) statement recommendations. The PRISMA Checklist is summarized in the Appendix A (see Appendix A), and the PRISMA statement is also summarized in the Appendix A (see Appendix A) [19]. The study protocol was determined prior to commencement, and it was registered in the PROSPERO-International prospective register of systematic review with the identification number CRD42016050563 available at: https://www.crd.york.ac.uk/prospero/#recordDetails.

### 2.2. Eligibility Criteria

Studies that assessed resistance in NTDs identified by the WHO were included in this review. All relevant studies were included irrespective of study design and countries of study. The included studies were limited to studies performed on human subjects only. Decisions on eligibility were made by two independent reviewers, all discrepancies and disagreements with respect to study and report eligibility were resolved through deliberations with a third reviewer.

### 2.3. Search Strategy

Studies analyzed in this review were identified by searching electronic public databases including: PubMed (http://www.ncbi.nlm.nih.gov/pubmed) and Scopus (http://www.scopus.com/). The searches were performed in December 2016 with a limit set for the dates of publications. The search focused on publications from 2000–2016. A full description of the search terms and search strategy is provided in Appendix A. Efforts were made to download the full text of the included articles, and when not available, the authors of such articles were contacted. For the unresponsive authors, reminders were sent to allow for a two-week period before such studies were excluded and classified as “full text not available”.

### 2.4. Study Selection

Initial eligibility assessments on the retrieved titles and abstracts were performed by two independent reviewers. Full texts of eligible articles were retrieved and reviewed for inclusion. The inclusion or exclusion of a study considered conclusive, controversial, or ambiguous by either of the reviewers was resolved through deliberations between the reviewers. When necessary, disagreements and discrepancies were resolved by consensus with a third reviewer. Adequate care was taken to identify more than one article reporting a single study. When this was encountered, the overlap was identified and resolved.

### 2.5. Assessment of the Methodological Quality

Based on the recommendations of a number of authors [20,21,22,23], the Quality Assessment Tool for Quantitative Studies (developed by the Effective Public Health Project) [22] was adapted for evaluating observational and experimental studies. The Assessment Tool contains 19 items in 8 key domains for evaluating study quality. The eight domains are study design, blinding, selection bias, withdraws/dropouts, confounders, data collection, data analysis, and reporting.

Using a range of 1 (low risk of bias; high methodological quality) to 3 (high risk of bias; low methodological quality), an overall rating for each study was determined based on their component rating. Strong was attributed to those with no weak ratings, and with at least five strong ratings. Moderate was assigned to those with one weak rating or fewer than five strong ratings. Finally, weak was attributed to those with two or more weak ratings. A full description of the Quality Assessment Tool for Quantitative Studies is provided in Appendix A.

All included studies were independently assessed for methodological quality by two assessors (FT and MN). The ratings for each of the eight key domains as well as the total rating from the two accessors were compared. Consensus was reached on a final rating for all included articles.

## 3. Results

Using the search terms for each identified major NTD, 1469 articles were screened based on their titles and abstracts, out of which 815 studies were selected for full-text reading. Based on the selected studies, two reviewers agreed on 145 decisions, and 37 discrepancies were resolved by discussion and consensus. A total of 108 studies were included in the final review (see Figure 1). The flowcharts for each reviewed NTD and the corresponding drugs for treatment are provided in Appendix A.

Out of 11 NTDs, 6 NTDs had information on AMR, namely HAT, leishmaniasis, onchocerciasis, schistosomiasis, soil-transmitted helminths, and trachoma. Out of a total of 108 studies, 79 were observational studies (26 cohort studies, 28 cross-sectional studies, 16 case reports, and 9 case-control studies), and 29 articles were experimental studies (21 random experimental studies, and 8 non-random experimental studies). The most studied NTDs were HAT and schistosomiasis. HAT had the highest number of cross-sectional studies, while schistosomiasis had the highest number of cohort studies. Study types with respect to the studied NTDs are presented in Figure 2.

HAT studies had 31% cross-sectional studies and 20% case reports. Schistosomiasis studies had 58% cohort studies and 32% cross-sectional studies.

Out of the 108 included studies, 70 studies were conducted in rural settings, 22 studies were conducted in urban settings, and 16 studies did not specify their study settings. Studies involving both genders were observed in 84 reviewed articles. Studies on males only were observed in 19 reviewed articles, and 5 studies were on females only. Reviewing the age range of the studies, 45 studies were conducted on both adults and children, 31 studies involved only adults, and 24 studies involved only children.

With respect to resistance of the reviewed studies, 92% of the articles indicated resistance due to diagnostic tests, while 42% of the studies indicated clinical resistance. This indicates a high resistance in both laboratory and clinical tests.

Out of the 28 countries of study, it was observed that most studies were performed in South Sudan, Tanzania, Kenya, Cameroon, Uganda, Cote d’Ivoire, the Democratic Republic of Congo (DRC), Angola, and India. There was no link between the countries of studies and disease burden as presented in Figure 3, Figure 4, Figure 5, Figure 6, Figure 7 and Figure 8.

Of the included studies, 78% were performed in Africa, 15% in Asia, and 1% in Australia.

Of the 84 studies conducted in Africa, 55% were on HAT, 20% on schistosomiasis, 8.3% on both onchocerciasis and trachoma, and 6.0% and 2.4% on soil-transmitted helminths and leishmaniasis, respectively. For Asia, a total of 17 studies were conducted, 47% on leishmaniasis, 18% on trachoma and soil-transmitted helminthiasis, 12% on schistosomiasis, and 5.9% on HAT. 

Detailed characteristics of the 108 studies included in this systematic review are summarized in Table 1. 

### Methodological Quality Assessment

In the overall assessment, the methodological quality of six reviewed studies were rated as strong, and 23 and 79 articles were rated as moderate and weak, respectively (see Appendix A for the full details of the quality assessment results). There were 20 reviewed studies rated as weak for their data collection because the authors did not provide sufficient information on the validity or reliability of their methods of collection. There were 40 articles rated as moderate, and 42 articles rated as strong. With respect to confounders, 37 articles were rated as weak, and 18 and 52 articles were rated as moderate and strong, respectively. Based on the data analysis of each reviewed study, 63.3% of the reviewed articles were rated as strong, while 14.3% and 22.4% were rated as moderate and weak, respectively. The reporting quality of the reviewed articles was also analyzed. Out of the 108 articles included, 59.3% were rated as strong, and 28.7% and 12.0% were rated as moderate and weak, respectively.

## 4. Discussion

This study reviewed 11 out of the 20 NTDs identified by the WHO as the most important NTDs [5] which have specific drugs for treatment. 

The major finding was that only six NTDs of those reviewed had information on AMR, namely, HAT, leishmaniasis, onchocerciasis, schistosomiasis, soil-transmitted helminths, and trachoma, while there was a deficiency of data to determine the magnitude and scope of resistance in the other reviewed NTDs. Moreover, it could be inferred that data deficiencies were from countries without surveillance or as a result of under-reporting in some countries. One of the main objectives of GAP-AMR is to ensure that there is the successful treatment and prevention of infectious diseases with effective and safe medicines that are quality assured and accessible to people at risk [15,25]. However, there is no harmonized system in place to standardize the collection of AMR data to facilitate a comprehensive purview of the global occurrence of AMR [14,15]. Even though PCT and MDA programs are currently effective in mitigating the morbidity of NTDs and improving the quality of life in the most affected countries [26], there is insufficient information on the program monitoring the effects of PCT and MDA. For AMR to be effectively and efficiently monitored, the collection of surveillance data is paramount to inform and estimate the AMR burden of NTDs. Following a recommendation by the Working Group on Monitoring and Evaluation of the Strategic and Technical Advisory Group for NTDs, the integrated NTD database developed by the WHO to improve the evidence-based planning and management of NTD programs at the national and sub-national levels does not contain information on AMR [27]. Immersed efforts are required to ensure that treatments are implemented efficiently and that monitoring and surveillance tools are improved [28]. It has been observed and argued that an effective surveillance-response system is a core feature of gaining reliable information on the prevalence, incidence, and burden of diseases which is essential for the prevention, control, and elimination of NTDs [16]. The health systems in countries where NTDs are endemic are often constrained by insufficient funding, limited human resources, insufficient management, and poor governance, which have impacts on the NTDs in these regions [29].

The number of studies that indicated resistance through diagnostic tests was considerably high in the reviewed studies. This indicates that accurate disease diagnoses are essential for appropriate and effective treatments, as well as disease control [30]. The Neglected Infectious Diseases Diagnosis consortium has been working to improve diagnostic approaches for different clinical syndromes in low-resource settings where NTDs are prevalent [30]. Moreover, this review indicates that clinical resistance was less than half in the studies. This suggests that observing people is not enough, and there is an urgent need for accessible diagnostic technologies for AMR [31]. It has been observed that the effect of PCT is weak compared to the original framework, and this may be as a result of increased drug pressure due to the mechanism of drug resistance. Drug efficacy monitoring is important for control programs based on PCT in order to support the correct use of antimicrobials (dosage, frequency, combinations), by ensuring the implementation of successful mitigation strategies [32,33]. Though pharmaceutical research and development had historically neglected the infectious diseases, there has been a recent increase in collaborative clinical research addressing the NTDs health needs of low-to-middle-income countries through therapeutic and diagnostic trials [34]. Furthermore, the Good Clinical Practices codes of the WHO and the International Conference of Harmonization provide globally applicable standards for designing, conducting, recording, and reporting clinical trials [34]. The respondents of a survey were of the opinion that drug resistance is a major challenge for the elimination of NTDs [35]. This opinion aligns with the experience of the global health community on malaria, in which resistance was found to have emerged after the mass distribution of medicated salts [36]. AMR has been recorded in malaria treatment, due to inappropriate, badly executed, or poorly accepted MDA. However, effective MDA with a good adherence routine can prevent the emergence of AMR [37]. The review findings also show that there was more information on the individual usage of the drugs compared to MDA. Studies on the PCT-MDA of NTDs like onchocerciasis, schistosomiasis, and soil-transmitted helminths had less information on their MDA programs. The issue of the low coverage of MDA programs is not a surprise, as it is a recognized challenge as the 2020 deadline for most NTDs approaches.

This review also indicates that some studies were conducted in countries where NTDs are prevalent but with less information on their AMR. Moreover, there are countries with high prevalence of NTDs (e.g., leishmaniasis is highly prevalent in Afghanistan, Yemen, Pakistan; onchocerciasis in DRC, and Nigeria), but there were no studies in this review performed in these countries. This might be as a consequence of either inefficient monitoring and surveillance tools or the political instability in these countries [38].

The overall description of the study settings of this review shows that most studies were conducted in the rural areas. Ponte-Sucre et al. highlighted that poor socioeconomic conditions are one of the fundamental contributory factors to AMR [26]. This concurs with the fact that these diseases are prevalent amongst the poorest populations of the world, putting an estimated 2.7 billion people at risk [39]. In spite of the international funding and support offered by the WHO and the existence of philanthropic organizations to fight NTDs in these at-risk poor populations, the public health burden and the challenges facing programs to achieve sustained control and ultimate elimination of major NTDs are still enormous [16]. Failure to tackle AMR threatens the attainment of various Sustainable Development Goals (SDGs), such as those on poverty reduction, reduced inequalities, clean water, economic growth, food security, and sanitation [18].

The burden of NTDs can be taken up by long-term capacity building and health-system-wide reforms, which will depend on health systems stepping up measures to meet the demands for services as part of their transition towards Universal Health Coverage (UHC). Therefore, there is much that NTD programs have to share with national health systems as they strive towards UHC [28], in order to improve monitoring and surveillance tools, coupled with effective reporting systems towards the progress 2020 roadmap targets for NTDs.

In identifying the resistance through a systematic review, data extraction and compilation are prone to bias. As a result, efforts were made to identify and screen published literature with a specific search query. Moreover, some relevant studies might have been excluded due to the search criteria narrowing publication dates from 2000–2016 due to both inaccessibility and lack of full text availability. Additionally, all studies with incomplete information were excluded. This review relied completely on published literature where grey literature and studies with minimal or negative results may not have been included, resulting in publication bias.

## 5. Conclusions

As 2020 approaches, it is essential to foster national surveillance systems and harmonize global standards that estimate the extent of AMR globally. The global community is committed to reducing the impact of NTDs on the poorest populations through the administration of PCT and medical care to people suffering from chronic manifestations of these NTDs. Presently, there is a global scale-up in PCT and drug donations through official development assistance and philanthropies, and hence there is an urgent need for effective and efficient data monitoring and national surveillance systems that will enable the early detection of AMR and the mitigation of its global spread. 

## Figures and Tables

**Figure 1 ijerph-16-01925-f001:**
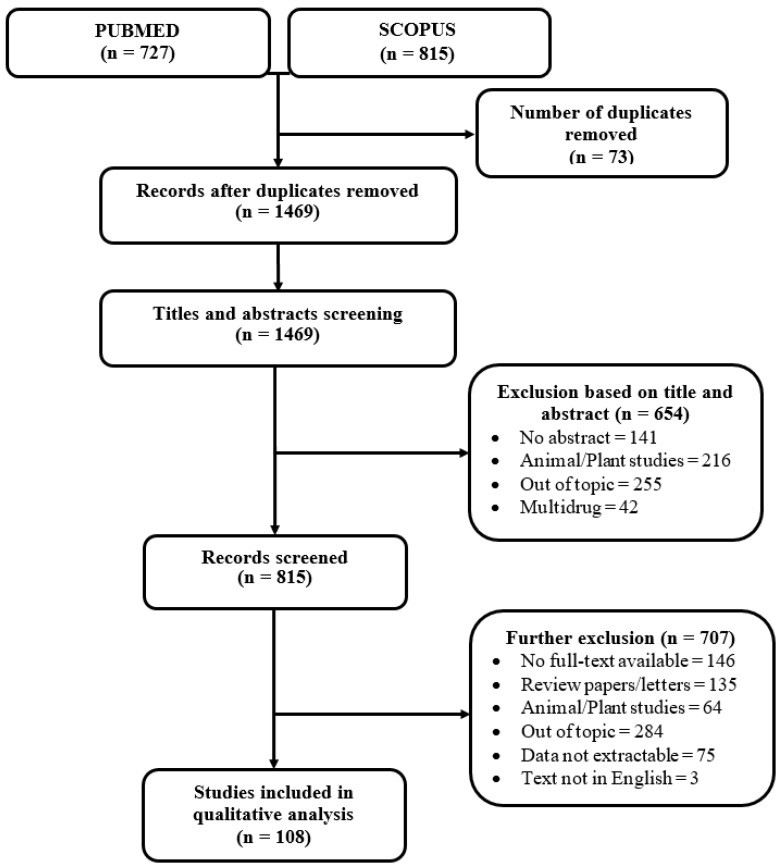
PRISMA flow chart.

**Figure 2 ijerph-16-01925-f002:**
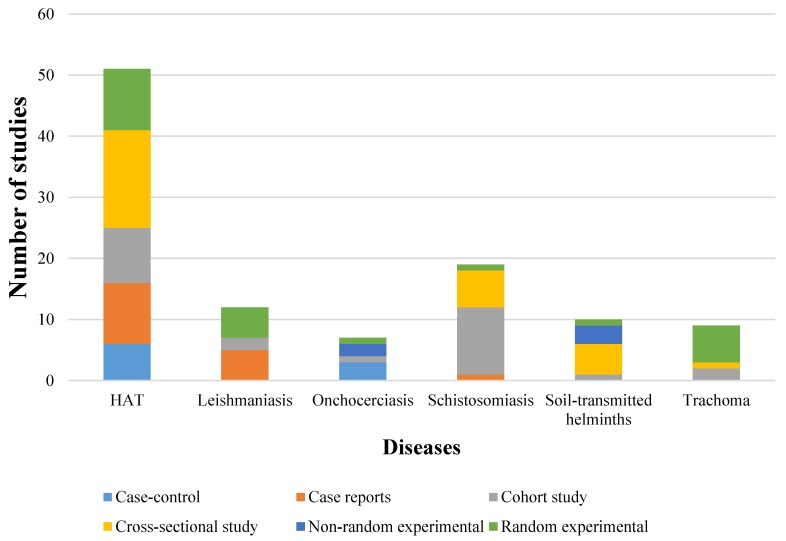
Prevalence of the study types of diseases. HAT: human African trypanosomiasis.

**Figure 3 ijerph-16-01925-f003:**
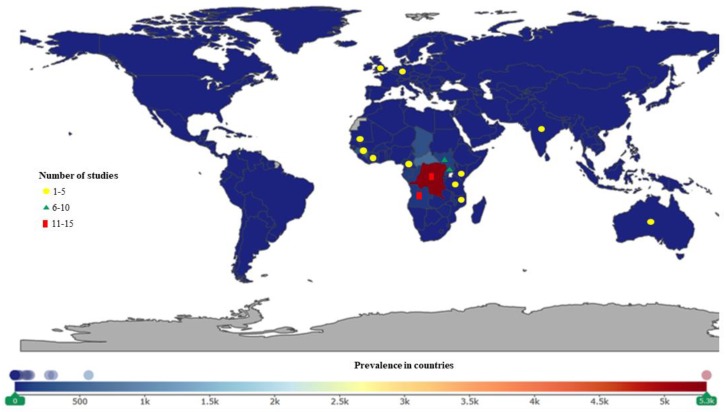
Human African trypanosomiasis (HAT) studies were conducted in Africa, Asia, Australia, and Europe. More studies were conducted in the Democratic Republic of Congo (DRC), Angola, South Sudan, and Uganda. The DRC, Central African Republic (CAR), and Gabon have the highest prevalence.

**Figure 4 ijerph-16-01925-f004:**
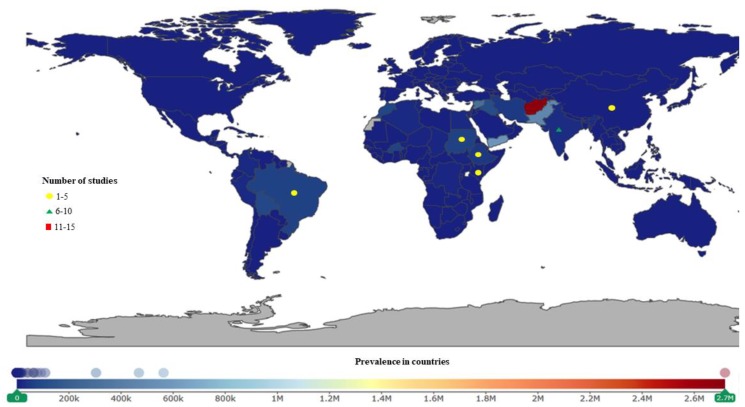
Leishmaniasis studies were conducted more often in Asia. India had the highest number of studies. Afghanistan, Yemen, and Pakistan have the highest prevalence.

**Figure 5 ijerph-16-01925-f005:**
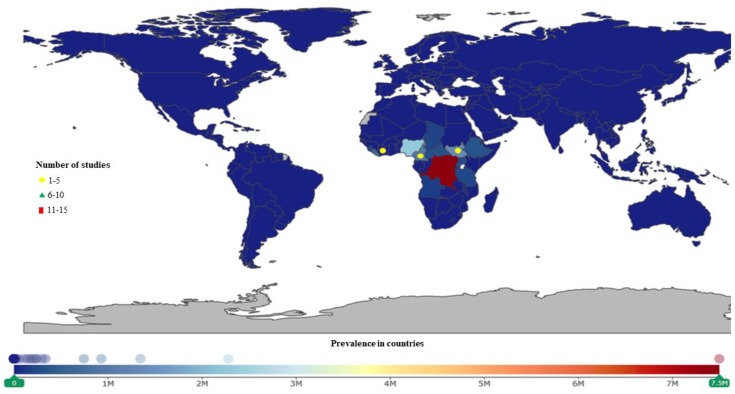
Soil-transmitted helminths studies were conducted more often in Africa. China, India, and Nigeria have the highest prevalence.

**Figure 6 ijerph-16-01925-f006:**
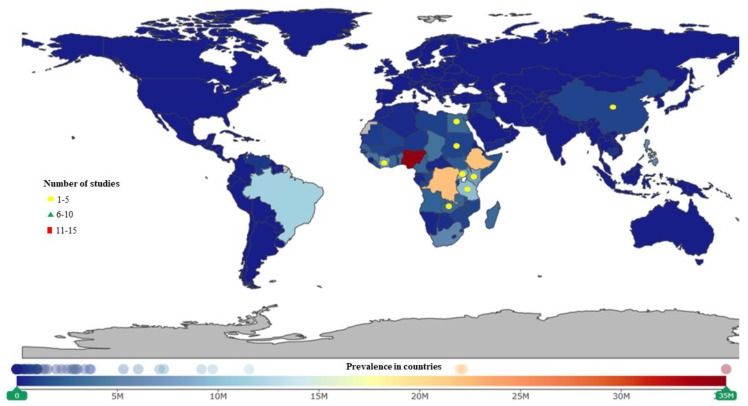
Schistosomiasis studies were mostly conducted in Africa. Egypt and Cote d’Ivoire had the most studies. Nigeria, the DRC, and Ethiopia have the highest prevalence.

**Figure 7 ijerph-16-01925-f007:**
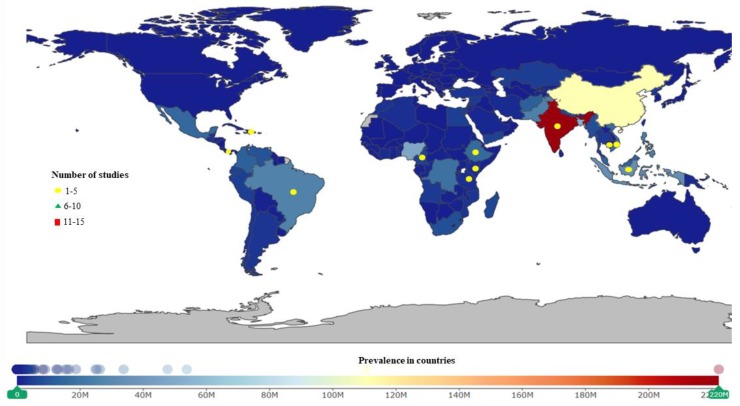
Trachoma studies were conducted more in Africa and Asia. Tanzania, Ethiopia, and Nepal had the most studies. Trachoma is highly prevalent in India, China, and Egypt.

**Figure 8 ijerph-16-01925-f008:**
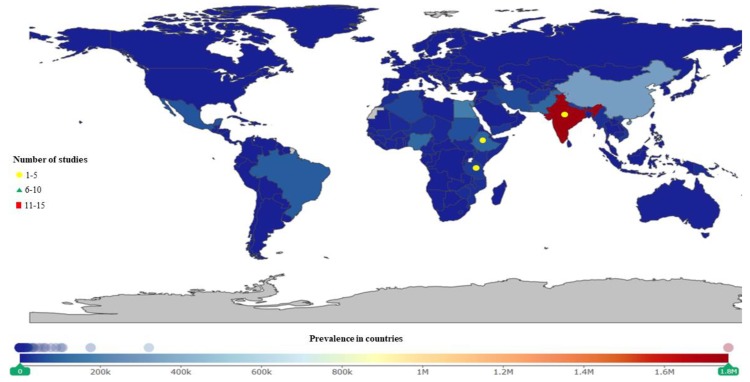
Onchocerciasis studies were conducted only in Africa (Cameroon, Ghana, and South Sudan). Onchocerciasis is highly prevalent in the DRC, Nigeria, and South Sudan. Source: The maps and prevalence of the studied neglected tropical diseases (NTDs) were created and retrieved from the Global Health Data Exchange [24].

**Table 1 ijerph-16-01925-t001:** The characteristics of reviewed NTDs and their treatments.

Data		Neglected Tropical Diseases (NTDs)									
		Human African Trypanosomiasis (Studied Drugs)				Leishmaniasis (Studied Drug)	Onchocerciasis (Studied Drug)	Schistosomiasis (Studied Drug)	Soil-Transmitted Helminths (Studied Drugs)		Trachoma (Studied Drug)
		Eflornithine	Melarsoprol	Pentamidine	Suramin	Amphotericin B	Ivermectin	Praziquantel	Albendazole	Mebendazole	Azithromycin
**Study Design**	Experimental	2	6	1	1	5	3	1	2	2	6
	Observational	8	14	12	7	7	4	18	3	2	4
**Study Settings**	Not Specified	4	3	3	2	0	0	1	2	0	1
	Rural	4	17	6	5	2	7	13	3	4	9
	Urban	2	0	4	1	10	0	5	0	0	0
**Gender**	Both	7	18	12	3	7	4	14	5	4	10
	Female only	1	0	0	0	1	0	3	0	0	0
	Male only	2	2	1	5	4	3	2	0	0	0
**Age Range**	Adults	6	8	3	5	5	4	6	0	0	0
	Children	0	0	1	1	1	0	7	5	3	8
	Both	4	12	8	2	6	3	6	0	1	2
**Resistance by Tests**	YES	9	18	13	8	9	6	18	5	4	9
	NO	1	2	0	0	3	1	1	0	0	1
**Clinical Resistance**	YES	6	6	7	7	10	0	4	1	0	4
	NO	64	14	6	1	2	7	15	4	4	6
**Countries of Study**		Angola, Cote d’Ivoire, Democratic Republic of Congo, Germany, South Sudan, Uganda, and Western Australia	Angola, Cameroon, Cote d’Ivoire, Central African Republic (CAR), Democratic Republic of Congo, Equatorial Guinea, Kenya, South Sudan, Tanzania, and Uganda	Angola, Central African Republic (CAR), Cote d’Ivoire, Democratic Republic of Congo, Equatorial Guinea, South Sudan, and Uganda	Belgium, Cameroon, Democratic Republic of Congo, India, England, Malawi and Tanzania	Brazil, China, Ethiopia, India, Sudan, and Kenya	Cameroon, Ghana, and South Sudan	China, Cote d’Ivoire, Egypt, Kenya, South Sudan, Tanzania, Uganda, and Zambia	Brazil, Cambodia, Cameroon, Ethiopia, Haiti, India, Indonesia, Kenya, Panama, Tanzania, and Vietnam	Indonesia and Tanzania	Ethiopia, Nepal, and Tanzania

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
