# Peer review of "Emerging Resistance of Neglected Tropical Diseases: A Scoping Review of the Literature"

_ijerph, 2019, doi:10.3390/ijerph16111925_

Round 1
Reviewer 1 Report
This review is a descriptive analysis of how many articles have examined resistance with respect to NTDs. The authors simple describe the count of articles and list the drugs investigated. The authors apply a methodological quality screening index/tool and note that only 6 NTD studies on resistance are of high quality.
The article would be better written as a scoping review that describes the estimated prevalence of drug resistance for different NTDs. The abstract claims to examine prevalence of resistance, but this information is absent from this manuscript.
Abstract
1. ‘Antimicrobial’ generally is not used to refer to parasites and, to my knowledge; I have yet to see this word used to describe resistance to preventive chemotherapy. In the case of preventive chemotherapy, only trachoma, which is treated through mass drug administration, is caused by bacteria.
2. Replace ‘antimicrobial’ with ‘drug’ or ‘medicine’ as appropriate, e.g. Line 10, Line 11, and so on. Replace throughout the entire manuscript.
3. Line 12: Delete ‘eradication.’ There are no preventive chemotherapy programmes targeting eradication.
4. Line 16: What does ‘a prior design’ mean?
5. Line 22: Test of what?
6. Lines 22-34: It is unclear what this sentence is trying to state.
7. Lines 24-26: This conclusion does not follow from the results summarized in the abstract. You do not state the 1) what the ‘prevalence of resistance’ refers to, 2) what the magnitude of that prevalence is, and 3) what impact this has on cure rates, etc.
Introduction
8. Lines 33-35: ‘Emergence’ is likely inappropriate here. Most NTDs are ancient diseases. Perhaps, you are referring to the high prevalence worldwide?
9. Lines 40-42: The term NTDs was developed by Peter Hotez, David Molyneux, and Alan Fenwick. Also, the publication referenced here [2] is inappropriate. It is a reference to developing patents, etc. The correct reference is Molyneux DH, Hotez PJ, Fenwick A. “Rapid-Impact Interventions”: How a Policy of Integrated Control for Africa's Neglected Tropical Diseases Could Benefit the Poor. PLOS Medicine 2005;2(11):e336. doi: 10.1371/journal.pmed.0020336.
10. Line 44: Remove reference [2]. Again, it is irrelevant and referring to patents, not the WHO report on NTDs.
11. Lines 45-47: Remove the quotes.
12. Lines 53-54: This statement is incorrect. MDA has been successful in reducing morbidity and prevalence.
13. Lines 54-56: A longer period than what? Please provide a justification for this statement. I suggest you turn to the NTD modeling consortium to view their publications.
14. Lines 57-59: Again, provide support for the statements that are made. Why is schistosomiasis and trachoma treatment inadequate?
Methods
15. Lines 109-114: Text is in a different font.
16. The methodological quality criteria need to be clearly explained.
Results
17. Line 151: Again, what is meant by ‘test’? Clarify.
18. Figure 3: The legends of this figure cannot be read. Separate into separate figures so the scale bar and legends are clear. Though, arguably, a table or bar chart showing the countries on the x-axis (e.g. for a bar chart) and the breakdown of diseases studied would be more useful.
19. Where are the results of socio-economic factors? Only a table with gender and age are provided. These are demographics.
20. What stage of the pathogen life cycle was tested or examined for resistance?
21. No prevalence information is provided. Simple counts do not equal prevalence. A population denominator is needed to understand the percentage of pathogens, people, etc. where resistance was found.
22. Only 6 NTDs are presented here (Table 1). I suggest the authors provide all studies of resistance and show their screening mechanisms (assigned scores) for methodological quality in a flowchart.
Author Response
Please read our responses to your comments and suggestions:
Abstract
1. ‘Antimicrobial’ generally is not used to refer to parasites and, to my knowledge; I have yet to see this word used to describe resistance to preventive chemotherapy. In the case of preventive chemotherapy, only trachoma, which is treated through mass drug administration, is caused by bacteria.RESPONSE: We appreciate your comment. We understood that, in case of preventive chemotherapy, only trachoma is caused by bacteria. Other NTDs in the study are caused by fungi and parasites. According to the WHO fact sheet on antimicrobial resistance (AMR), AMR is defined as resistance that happens when microorganisms (such as bacteria, fungi, viruses, and parasites) change when they are exposed to antimicrobial drugs (such as antibiotics, antifungals, antivirals, antimalarial, and anthelmintic).
Reference:
a) World Health Organization. Fact Sheets on Antimicrobial resistance.(https://www.who.int/en/news-room/fact-sheets/detail/antimicrobial-resistance)
b) World Health Organization. Global antimicrobial resistance surveillance system ( GLASS) report: early implementation 2017-2018.
2. Replace ‘antimicrobial’ with ‘drug’ or ‘medicine’ as appropriate, e.g. Line 10, Line 11, and so on. Replace throughout the entire manuscript.
Response: Thank you for your suggestion, however, to our knowledge, “antimicrobial” is valid for an agent that kills microorganisms or stops their growth.
Reference:
Review on Antimicrobial Resistance. Tackling drug-resistant infections globally: final report and recommendations. Review on antimicrobial resistance; 2016.
3. Line 12: Delete ‘eradication.’ There are no preventive chemotherapy programmes targeting eradication.
Response: The word ‘Eradication’ has been removed from the sentence. The new sentence states that, “There is a tremendous increase of antimicrobials use stemming from preventive chemotherapy elimination and control programs addressing neglected tropical diseases (NTDs).”
4. Line 16: What does ‘a prior design’ mean?
Response: The term ‘a prior design,’ explains that the review was carried out based on an existing study protocol which was designed to identify publications on drug resistance. It has been removed to avoid confusion.
5. Line 22: Test of what?
Response: For clarity, the sentence has been changed in the manuscript to “a high number of studies recorded resistance due to diagnostic tests, and few studies indicated clinical resistance.”
6. Lines 22-34: It is unclear what this sentence is trying to state.
Response: Thank you for your comment. The sentence has been rephrased to “Although, most studies were performed in Africa where there is an occurrence of several NTDs, there is no link between disease burden and locations of study.”
7. Lines 24-26: This conclusion does not follow from the results summarized in the abstract. You do not state the 1) what the ‘prevalence of resistance’ refers to, 2) what the magnitude of that prevalence is, and 3) what impact this has on cure rates, etc.
Response: Since our sentences on ‘prevalence’ were misleading. The sentences have been rephrased in the entire manuscript. More also, our study strongly emphasized on the effectiveness of monitoring and surveillance systems for NTDs.
Introduction
8. Lines 33-35: ‘Emergence’ is likely inappropriate here. Most NTDs are ancient diseases. Perhaps, you are referring to the high prevalence worldwide?
Response: Thank you, the sentence has been revised.
The widespread and often catastrophic consequences of NTDs necessitates a global response, prompting organizations such as United States Centers for Disease Control and Prevention (CDC), United Nations (UN) and World Health Organization (WHO) to focus on them.
9. Lines 40-42: The term NTDs was developed by Peter Hotez, David Molyneux, and Alan Fenwick. Also, the publication referenced here [2] is inappropriate. It is a reference to developing patents, etc. The correct reference is Molyneux DH, Hotez PJ, Fenwick A. “Rapid-Impact Interventions”: How a Policy of Integrated Control for Africa's Neglected Tropical Diseases Could Benefit the Poor. PLOS Medicine 2005;2(11):e336. doi: 10.1371/journal.pmed.0020336.
Response: Thank you for your observation, the reference has been changed to the original one.
10. Line 44: Remove reference [2]. Again, it is irrelevant and referring to patents, not the WHO report on NTDs.
Response: The reference [2] has removed from the section.
11. Lines 45-47: Remove the quotes.
Response: Thank you, the quotes have been removed.
The two primary methods of interventions for NTDs are preventive chemotherapy and transmission control (PCT) which covers mass drug administration (MDA), and innovative and intensified disease management (IDM).
12. Lines 53-54: This statement is incorrect. MDA has been successful in reducing morbidity and prevalence.
Response: Thank you for your observation, the statement has been rephrased to“In 2011, there was a 37% average coverage of PCT for NTDs but with the involvement of strong partnerships, the average coverage of PCT increased to 63% in 2016. PCT covering MDA programs has been considered to be effective, in spite of the potential for development of drug resistance due to long and continuous usage that remains a challenge.”
13. Lines 54-56: A longer period than what? Please provide a justification for this statement. I suggest you turn to the NTD modelling consortium to view their publications.
Response: The statement has been rephrased to avoid confusion. Please see my previous response (12).
14. Lines 57-59: Again, provide support for the statements that are made. Why is schistosomiasis and trachoma treatment inadequate?
Response: An additional reference has been added to support our statement. According to WHO, less than one-third of populations requiring trachoma treatments and less than one-half of school-aged children needing treatment for schistosomiasis or preschool-aged children needing deworming for their soil transmitted helminthiases has been reached compared to 60% of the global population that have been reached for lymphatic filariasis, onchocerciasis and soil-transmitted helminth.
Reference:
a) Hotez PJ. Ten failings in global neglected tropical diseases control
(https://www.ncbi.nlm.nih.gov/pmc/articles/PMC5739381/#pntd.0005896.ref01)
b) WHO. Summary of global update on preventive chemotherapy implementation in 2015 Preventive chemotherapy (PC) is referred. Weekly Epidemiol Rec. 2016;91(39):441-60.
Methods
15. Lines 109-114: Text is in a different font.
Response: The text has been reformatted to merge the manuscript original format
16. The methodological quality criteria need to be clearly explained.Response: Thank you for your comment. A methodological quality criteria tool has been as a Supplementary Appendix 4 (full details of the quality assessment tool). Below is the quality assessment tool:
QUALITY ASSESSMENT TOOL
STUDY # RATER __________
STUDY DESIGN
(Q1) The study design is:
1. Experimental
i. Individual-randomised
ii. Group-randomised
iii. Non-randomised
2. Observational
i. Cross-sectional
ii. Longitudinal (also natural experiment or pre-post-tests)
iii. Case-control
3. Any other method or did not state method (i.e. pre-post-test without control group)
(Q2) Was this an intervention study?
Yes – proceed
No – go to question 7
(Q3) Is the intervention of interest clearly described?
1. Yes
2. No
(Q4) Were (groups of) subjects randomized into intervention groups?
1. Yes
2. No
3. Not applicable
(Q5) Was the intervention assignment concealed from participants and care givers until recruitment was completed?
1. Yes
2. No
3. Can’t tell
(Q6) Was (were) the intervention or exposure status of participants concealed from the outcome assessors?
1. Yes
2. No
3. Can’t tell
(Q7) Were power/sample size calculations conducted?
1. Yes, details of calculation provided
2. Yes, no details provided
3. Not reported or post hoc computation
4. Not applicable (using an existing database and referring to design article*
Rating study design: Strong: Q1 is 1
Moderate: Q1 is 2
Weak: Q1 is 3
Rating blinding: Strong: Q5 and Q6 are 1
Moderate: Q5 or Q6 is 1; or Q5 or Q6 are 3
Weak: Q5 and Q6 are 2; or Q5 and Q6 are 3
(No rate is given when study is not an intervention study)
* If the study is using data from a large existing database such as HSE, NHANES, BRFSS etc, often the authors refer to the design paper of the original study and no information in the present article is being described about power calculations, validity of tools et.
REPRESENTATIVENESS (selection bias)
(Q8) Is the spectrum of individuals selected to participate likely to be representative of the wider population who experience the intervention/exposure/situation?
1. Very likely
2. Somewhat likely
3. Not likely (selected group of users e.g., volunteers)
4. Can´t tell (no information provided)
5. Not applicable (using an existing database and authors refer to design article)
(Q9) What percentage of the selected participants agreed to participate?
…………..%
Can’t tell
Not applicable
(Q10) Were inclusion/exclusion criteria specified and number of exclusions reported?
1. Criteria and number of exclusions reported
2. Criteria or number of exclusions not reported
3. Criteria and number not reported
Rating: Strong: Q8 is 1
Moderate: Q8 is 2
Weak: Q8 is 3 or 4
No rating: Q8 is 5
REPRESENTATIVENESS (withdrawals and drop-outs)
(Q11) Were withdrawals and drop-outs reported in terms of numbers and reasons per group?
1. Numbers and reasons provided
2. Numbers but no reasons provided
3. Can’t tell (if longitudinal data)
4. Not applicable (if cross-sectional data or if using an existing database and authors refer to design article)
If Q11 is 1 or 2, proceed to Q12. Otherwise, proceed to Q13.
(Q12) What was the loss to follow-up/percentage completing the study? (If % differs by groups, record the lowest)
…………..%
Not provided
Not applicable
Rating: Strong: Q11 is 1
Moderate: Q11 is 2
Weak: Q11 is 3
No rating: Q11 is 4
CONFOUNDERS
(Q13#) What confounders were the analyses adjusted for?
…………………………………………………………………………………………………..
…………………………………………………………………………………………………..
(Q13) Were analyses appropriately adjusted for confounders?
1. For most confounders
2. For some confounders
3. No or can’t tell
The following are examples of confounders: race, sex, marital status/family, age, SES (income or class), education, health status, and pre-intervention score on outcome measure.
Considering the study design, were appropriate methods for controlling confounding variables and limiting potential biases used? Confounding can be addressed by appropriate use of randomization, restriction, matching, stratification, or multivariable methods. Sometimes use of a single method may be inadequate. Some biases can be limited by institution of data collection or study procedures that support validity of the study (e.g. training and/or blinding of interviewers or observers, interviewers and observers are different from interventions’ implementers etc). Example: if between-group differences persist after randomization or matching, statistical control should also have been used.
Rating: Strong: Q13 is 1
Moderate: Q13 is 2
Weak: Q13 is 3
DATA COLLECTION
(Q14) Were validity, reliability or appropriateness of the data collection tools discussed?
1. Both validity and reliability were discussed
2. a. Validity or reliability were discussed
b. A national dataset was used and authors provided adequate information to find information on validity and reliability
3. None of them were discussed
Rating: Strong: Q14 is 1
Moderate: Q14 is 2
Weak: Q14 is 3
DATA ANALYSIS
(Q15) Were appropriate statistical analyses conducted (including correction for multiple tests where applicable)?
1. Statistical methods were described and were appropriate and comprehensive
2. Statistical methods were described and less appropriate
3. No description of statistical methods or inappropriate methods
Rating: Strong: Q15 is 1
Moderate: Q15 is 2
Weak: Q15 is 3
REPORTING
(Q16) Are the hypothesis/aim/objective of the study clearly described?
1. Yes
2. No
(Q17) Are the main outcomes to be measured clearly described?
1. Yes
2. No
(Q18) Are the main findings clearly described?
1. Yes
2. No
(Q19) Have actual probability values been reported?
1. Yes
2. No
Rating: Strong: Q16 and Q19 are 1
Moderate: Q16 or Q19 are 1
Weak: Q16 and Q19 are 2
Studies can have between six and eight component ratings. The overall rating for each study is determined by assessing the component ratings.
If eight ratings have been given;
Strong will be attributed to those with no weak ratings and at least five strong ratings; Moderate will be given to those with one weak rating or fewer than five strong ratings; Weak will be attributed to those with two or more weak ratings.
If seven ratings have been given;
Strong will be attributed to those with no WEAK ratings and at least four STRONG ratings; Moderate will be given to those with one WEAK rating or fewer than four STRONG ratings; Weak will be attributed to those with two or more WEAK ratings.
If six ratings have been given;
Strong will be attributed to those with no WEAK ratings and at least three STRONG ratings; Moderate will be given to those with one WEAK rating or fewer than three STRONG ratings;
Weak will be attributed to those with two or more WEAK ratings.
If five ratings have been given;
Strong will be attributed to those with no WEAK ratings and at least two STRONG ratings; Moderate will be given to those with one WEAK rating or fewer than two STRONG ratings;
Weak will be attributed to those with two or more WEAK ratings.
If four ratings have been given;
Strong will be attributed to those with no WEAK ratings and at least two STRONG ratings; Moderate will be given to those with one WEAK rating or fewer than two STRONG ratings;
Weak will be attributed to those with two or more WEAK ratings.
The final decision of both reviewers will be: strong, moderate, or weak
Results
17. Line 151: Again, what is meant by ‘test’? Clarify.
Response: Thank you for your comment. The sentence has been rephrased to “With respect to resistance of the reviewed studies, 92% of the articles indicated resistance due to diagnostic tests while 42% of the studies indicated clinical resistance. This indicates a high resistance in both laboratory and clinical tests.”
18. Figure 3: The legends of this figure cannot be read. Separate into separate igures so the scale bar and legends are clear. Though, arguably, a table or bar chart showing the countries on the x-axis (e.g. for a bar chart) and the breakdown of diseases studied would be more useful.
Response: The figures have been separated for better resolution.
19. Where are the results of socio-economic factors? Only a table with gender and age are provided. These are demographics.
Response: Thank you for your comment. Our study aimed to analyse socio-demographic factors only. A reference has been added to justify our meaning of socio-demographics, being gender, ethnicity, age, relationship status and place of residence.
Reference:
Hatch SL, Frissa S, Verdecchia M, Stewart R, Fear NT, Reichenberg A, Morgan C, Kankulu B, Clark J, Gazard B, Medcalf R. Identifying socio-demographic and socioeconomic determinants of health inequalities in a diverse London community: the South East London Community Health (SELCoH) study. BMC Public Health. 2011 Dec;11(1):861.
20. What stage of the pathogen life cycle was tested or examined for resistance?
Response: This study has no limitation for the pathogen life cycle.
21. No prevalence information is provided. Simple counts do not equal prevalence. A population denominator is needed to understand the percentage of pathogens, people, etc. where resistance was found.
Response: We apologize for using the term “prevalence.” It has removed from the manuscript. The study is a scoping review, we relied on the information provided by each study that recorded resistance on humans.
22. Only 6 NTDs are presented here (Table 1). I suggest the authors provide all studies of resistance and show their screening mechanisms (assigned scores) for methodological quality in a flowchart.
Response: All the studies of resistance and their screening mechanisms are available as a Supplementary Appendix 5 (Quality assessment of included studies).
Thank you for your suggestion, the title of the manuscript has been changed to “Emerging Resistance of Neglected Tropical Diseases: A scoping review of the literature”

Reviewer 2 Report
General comments
This is an interesting manuscript, bringing an extensive revision focusing on the emergence (or stablished) drug resistance for major tropical diseases, highlighting the absence of clear data about this threat, even in some parasitic disease under mass drug control. There is no doubt that pharmacoresistance is of major concern, consideration the lack of alternative and effective antiparasitic drugs for the most NTDs, therefore it is important to provide accurate information’s, as discussed in this MS, in order to prevent or minimize the spread of drug resistance.
Overall, the MS is clear and well written and the selective methodology seems adequate for this kind of research, based on results of field studies performed with so distint protocols. I recommend the MS publication.
Author Response
Thank you for reviewing our manuscript titled “Emerging Resistance of Neglected Tropical Diseases: A systematic review of the literature.” We sincerely appreciate your constructive reviews and recommendation for publication.

Reviewer 3 Report
Line 26 of abstract, stray period. .
Line 109-114: Text seems larger or different font here
Line 109: Delete ‘good’
Line numbers stopped after Table 1 but in the discussion ‘It has been observed that PCT suboptimal effect is weak’ – seems like there might be a word missing? Doesn’t quite make sense
Nice paper, and obviously a very important topic. MDA seems to be the norm for NTD control, and it provides the perfect conditions for resistance to occur.
Author Response
Please read our responses to your comments and suggestions:
1. Line 26 of abstract, stray period. .
Response: Thank you for your observation, the stray period has been deleted.
2. Line 109-114: Text seems larger or different font here
Response: The text has been reformatted to merge the manuscript original format.
3. Line 109: Delete ‘good’
Response: The ‘good’ has been deleted.
“Based on the recommendations of a number of authors……….”
4. Line numbers stopped after Table 1 but in the discussion ‘It has been observed that PCT suboptimal effect is weak’ – seems like there might be a word missing? Doesn’t quite make sense.
Response: Thank you for your comment. The sentence has been rephrased to
“It has been observed that PCT effect is weak compared to the original framework, and this maybe as a result of increased drug pressure due to the mechanism of drug resistance.”
Reviewer 4 Report
This study deals with a very important topic from a public health and health policy point of view. It covers one and a half decades, focuses on antimicrobial resistance (AMR) in neglected tropical diseases (NTDs) mostly spread in African countries and provides conclusions based on a meta-analysis of a number of carefully selected researches.
Indeed, the challenge is to create cost-effective, accurate, rapid and easy-to-use tests for NTDs that allow health professionals worldwide to administer the right antibiotics at the right time. It is a demanding job, particularly in low-income countries where the public health infrastucture is weak and the uncontrolled antibiotic consumption is prevalent.
This situation is well illustrated by the major finding of the study, namely that only 6 NTDs out of the 11 reviewed NTDs has information on AMR. Although the discussion part as well as the brief conlusion section contain epidemiologically correct statements, even highlight the vulnerability of the poorer groups, it would be worthwhile to draw attention - beyond increasing antimicrobial awareness - to the the need of better addressing the problem in agriculture (the use of sub-therapeutic doses of antibiotics in animal rearing). In addition, more cooperation and investment are unavoidable in the pharmaceutical industry. The authors could make a reference to the health related Sustainable Development Goals (SDGs) by stating that AMR can compromise the achievement of the SDGs, affecting health security, poverty, economic growth and food security.
As WHO suggests, political decisions are needed to stop the unfavorable trends in the context of NTDs/AMR supported by effective surveillance in antimicrobial consumption and resistance in agriculture and veterinary sectors, the implementation of animal immunization and the promotion of improved hygiene and biosecurity.
Minor remarks:
On page 10, UHC should be interpreted as Universal Health Coverage (not as Universal Health Care).
In the concluding part, speaking about "worldwide scale-up in PC and drug donations", there is a need to clarify what PC means here (does it mean PCT?). Re drug donations: one may add "through the channels of ODA (official development assistance) and philantropies".
Author Response
Please read our responses to your comments and suggestions:
1. This situation is well illustrated by the major finding of the study, namely that only 6 NTDs out of the 11 reviewed NTDs has information on AMR. Although the discussion part as well as the brief conclusion section contain epidemiologically correct statements, even highlight the vulnerability of the poorer groups, it would be worthwhile to draw attention - beyond increasing antimicrobial awareness - to the need of better addressing the problem in agriculture (the use of sub-therapeutic doses of antibiotics in animal rearing). In addition, more cooperation and investment are unavoidable in the pharmaceutical industry.
Response: Thank you for your suggestion. The study focuses on human subjects only, that is the reason our discussion was on the monitoring and surveillance of AMR in humans. Not with standing, we have included the challenges in addressing AMR in agriculture and animal health in the introduction as stated below:
“Importantly, AMR does not recognize geographic or human/animal species borders. Addressing the rising threat of AMR requires the “One Health” approach, which addresses human health, animal health, and the environment. Although, the WHO, the Food and Agriculture Organization and the World Organisation for Animal Health have taken collective action to minimize the emergence and spread of AMR through the “Tripartite Collaboration”, but there are still limitations with the agreement. A collective action is required in areas of surveillance, infection control, awareness, and responsible use for successful containment of AMR emergence and spread.”
2. The authors could make a reference to the health related Sustainable Development Goals (SDGs) by stating that AMR can compromise the achievement of the SDGs, affecting health security, poverty, economic growth and food security.
Response: This statement has been included in the discussion.
“Failure to tackle AMR threatens the attainment of various SDGs such as those on poverty reduction, reduced inequalities, clean water, economic growth, food security and sanitation.”
3. On page 10, UHC should be interpreted as Universal Health Coverage (not as Universal Health Care).
Response: Thank you for your observation, UHC has been changed to “Universal Health Coverage”
4. In the concluding part, speaking about "worldwide scale-up in PC and drug donations", there is a need to clarify what PC means here (does it mean PCT?). Re drug donations: one may add "through the channels of ODA (official development assistance) and philanthropies".
Response: The new statement is: “Presently, there is a worldwide scale-up in PCT and drug donations through official development assistance (ODA) and philanthropies, hence, there is an urgent need for effective and efficient data monitoring and national surveillance systems that will enable early detection of AMR and the mitigation of its global spread.”

Round 2
Reviewer 1 Report
Fine